# Bioactive Peptides against Human Apicomplexan Parasites

**DOI:** 10.3390/antibiotics11111658

**Published:** 2022-11-19

**Authors:** Norma Rivera-Fernández, Jhony Anacleto-Santos, Brenda Casarrubias-Tabarez, Teresa de Jesús López-Pérez, Marcela Rojas-Lemus, Nelly López-Valdez, Teresa I. Fortoul

**Affiliations:** 1Departamento de Microbiología y Parasitología, Facultad de Medicina, Universidad Nacional Autónoma de México (UNAM), Ciudad Universitaria, Mexico City 04510, Mexico; 2Departamento de Biología Celular y Tisular, Facultad de Medicina, Universidad Nacional Autónoma de México (UNAM), Ciudad Universitaria, Mexico City 04510, Mexico; 3Posgrado en Ciencias Biológicas, Universidad Nacional Autónoma de México (UNAM), Ciudad Universitaria, Mexico City 04510, Mexico

**Keywords:** Apicomplexan, bioactive peptides, toxoplasmosis, cryptosporidiosis, malaria

## Abstract

Apicomplexan parasites are the causal agents of different medically important diseases, such as toxoplasmosis, cryptosporidiosis, and malaria. Toxoplasmosis is considered a neglected parasitosis, even though it can cause severe cerebral complications and death in immunocompromised patients, including children and pregnant women. Drugs against *Toxoplasma gondii*, the etiological agent of toxoplasmosis, are highly toxic and lack efficacy in eradicating tissue cysts, promoting the establishment of latent infection and acute relapsing disease. Cryptosporidiosis has been recognized as the most frequent waterborne parasitosis in US outbreaks; anti-cryptosporidium drug discovery still faces a major obstacle: drugs that can act on the epicellular parasite. Severe malaria is most commonly caused by the progression of infection with *Plasmodium falciparum*. In recent years, great progress has been made in the field of antimalarial drugs and vaccines, although the resistance of *P. falciparum* to artemisinin has recently gained a foothold in Africa. As seen, the search for new drugs against these parasites remains a challenge. Peptide-based drugs seem to be attractive alternative therapeutic agents recently recognized by the pharmaceutical industry, as they can kill different infectious agents and modulate the immune response. A review of the experimental effects of bioactive peptides on these parasites follows, along with comments. In addition, some biological and metabolomic generalities of the parasites are reviewed to elucidate peptide mechanisms of action on Apicomplexan targets.

## 1. Introduction

Parasitism is a biological interaction present in nature. Some parasites can cause a severe clinical picture, and others can even cause host death. Millions of people are infected by parasites worldwide, mainly in lower- and middle-income countries. Among the most important human parasites are single-cell protozoan organisms, which are divided into different phyla [1,2]. The protozoan phylum Apicomplexa is a large group of intracellular alveolates; its name is derived from the complex of organelles located at the apical end that allow them to survive in the host cell. Apicomplexan parasites cause important infectious diseases in humans, including malaria, toxoplasmosis, and cryptosporidiosis [3]. Some intestinal coccidian infections and toxoplasmosis are considered by the World Health Organization (WHO), neglecting parasitosis; therefore, they are not a priority for pharmaceuticals to invest in the research of new compounds for their control, and malaria is one of the most dangerous infections that caused approximately 627,000 human deaths in 2020 [4]. Anti-*Toxoplasma* drugs are highly toxic and ineffective in destroying tissue cysts, and cryptosporidiosis treatments are partially effective mostly in immunocompromised patients. Despite antimalarial drug research on the development of novel treatments, the emergence of strains resistant to first-line drugs is increasing; therefore, new alternatives are necessary [5,6]. Based on this background, a search for active molecules is needed. Drug development against these parasites has been approached from different perspectives, including in silico models, hybrid compound design, bio-guided studies in natural products, and even the use of combined therapies with known antibiotic drugs [7].

An interesting emerging category of active molecules is antimicrobial peptides (AMPs), which are attractive alternative therapeutic agents. Peptides are a diverse group of proteins of 10–100 amino acid residues. They have amphipathic structures, contain up to 50% hydrophobic residues, and possess a net positive charge of +2 to +9 [8]. AMPs are found naturally in tissues and cells from multicellular organisms and play a crucial role in the innate immune response to protect themselves since these organisms do not develop an adaptive immune system such as vertebrates. The interest in these compounds is due to their biochemical features that can interfere with ion channels and structural components of the cell membrane [9,10]. The first AMP was identified in mid-1990 from *Drosophila melanogaster;* at the time of this writing, at least 5000 AMPs have been reported [11,12].

The applications of AMPs are still under constant investigation, and in the last decade, their interesting antibacterial drug resistance, anticancer, anti-inflammatory, immunomodulatory, and antiparasitic activities have been reported [13,14,15]. However, the clinical application of AMPs has been limited due to the toxicity and stability of these molecules and other drawbacks, such as high production costs compared to conventional antibiotics. Although there are no commercial AMP products to date, we cannot ignore the great potential of AMPs. These molecules offer great alternatives due to their results in in vitro models [16,17].

In this review, we provide an in-depth overview of the main Apicomplexan human parasites and AMPs with antiparasitic activity, as well as their mechanisms of action.

### 1.1. Toxoplasmosis

This parasitic infection is caused by *Toxoplasma gondii*, an obligate intracellular distributed worldwide that infects a wide range of homothermic animals, including humans [18,19]. It is recognized as the main public health problem in human and veterinary medicine and is one of the five neglected parasitic infections cited by the WHO. *T. gondii* sexual reproduction involves species from the Felidae family, including domestic cats [20]. *T. gondii* affect approximately one-third of the human population, and climate change is increasing its prevalence of infection [21,22]. Epidemiological studies worldwide revealed that the prevalence in pregnant women is approximately 1.1% and could be related to cultural habits, such as eating undercooked meat (one of the main risk factors for *T. gondii* infection), especially of pork, lamb, or venison [23,24,25]. Humans can also be infected by eating raw shellfish (like oysters, clams, and mussels), by accidental ingestion of oocysts in contaminated soil, or by congenital transmission [25].

The toxoplasmosis incubation period is 10 to 14 days, and 90% of cases are asymptomatic. In symptomatic individuals, lymphadenitis, lymphadenopathy, fever, sore throat, headache, and myalgia have been reported [26]. The presence of hepatosplenomegaly, pulmonary or cardiac symptoms, conjunctivitis, and skin rash were recorded. Clinical manifestations are generally self-limited within 3–4 weeks. In immunocompetent individuals, neurological symptoms rarely occur; in some exceptional cases, moderate cognitive impairment has been reported [26]. In immunocompromised people with toxoplasmosis, parasites have a predilection for immune privilege sites, and extensive cell lesions are present, which can lead to encephalitis, retinochoroiditis, pericarditis, interstitial pneumonia, and Guillain-Barre syndrome. Encephalitis is an important clinical manifestation, especially in patients with AIDS, and congenital infections can lead to death [27].

In the biological life cycle of *T. gondii*, four parasitic forms are involved: tachyzoites, bradyzoites, tissue cysts, and oocysts. Definitive hosts ingest prey infected with tissue cysts, mainly in the skeletal muscle or brain. Due to digestive action, the bradyzoites contained in the tissue cysts invade the enterocytes and, through schizogony replication, differentiate into macro- and microgametes. Subsequently, fertilization takes place, which gives rise to a zygote. This zygote transforms into an immature, noninfectious oocyst that is released into the environment along with the host’s feces. The noninfecting oocyst sporulates and becomes infective, and contaminates water, soil, and food in favorable environmental conditions. Intermediate hosts (i.e., warm-blooded animals, including humans) become infected through the consumption of water and food contaminated with sporulated oocysts or raw or undercooked meat with tissue cysts. Oocysts and tissue cysts release sporozoites and bradyzoites, respectively, and differentiate into tachyzoites within the intestinal epithelium. After replication, the tachyzoites exit the cell, destroying it, and the infection spreads to neighboring cells. The immune response will eliminate most parasites; those that are not removed will become bradyzoites and will form tissue cysts that can remain in the host’s organs and tissues throughout life (chronic infection). In immunocompromised individuals, bradyzoites differentiate back to tachyzoites, causing severe or fatal acute disseminated infection [18,28,29] (Figure 1).

A combination of dihydrofolate reductase inhibitors such as pyrimethamine and trimethoprim, and dihydropteroate synthetase inhibitors (sulfonamides) are currently used as the first-choice treatment for toxoplasmosis; nevertheless, drug-resistant strains have been reported. It is worth mentioning that in the last decade, more than 50 resistant strains were identified and have developed resistance mainly to sulfonamides [30,31]. In addition to this, the presence of adverse effects and the fact that treatments are only effective in the acute phase of infection, turn out necessary to have new alternatives to treatment that are safe, effective, affordable, and active against the tissue cysts. For this reason, the recent emergence of AMPs offers wide potential for the discovery of new anti-*Toxoplasma* drugs. In Figure 2, drugs that have been tested against *Toxoplasma* are described.

### 1.2. Cryptosporidiosis

*Cryptosporidium* spp. is an important public health problem currently recognized as the main cause of diarrhea in humans and farm animals, causing significant morbidity and mortality worldwide, mainly in children. Approximately 40 species have been described in the *Cryptosporidium* genus. Two species are the most common, *Cryptosporidium hominis* and *C. parvum*, both of which can infect humans. *C. parvum* also infects cattle [32,33]. In low-income countries, 54% of children have had diarrhea associated with cryptosporidiosis. Children and immunocompromised patients are the most vulnerable groups to *Cryptosporidium* infections. It is estimated that two million children die worldwide annually, and 7 million cases are associated with morbidity in Asian and African populations [34]. In the last seventeen years, the incidence of *Cryptosporidium* infection in HIV-positive patients has increased up to 41.3% in Russia [35].

*Cryptosporidium* incubation period takes a week after the ingestion of infective oocysts. The clinical manifestations include diarrhea, fever, nausea, vomiting, abdominal pain, general malaise, and malnutrition. Chronic diarrhea in HIV patients is recognized as a classical clinical manifestation, and severe dehydration, weight loss, and malnutrition that can lead to death have been observed [36,37].

There are different parasitic stages in the life cycle of *Cryptosporidium* spp.: oocysts, sporozoites, trophozoites, and merozoites. The oocyst is the infective stage and can be consumed in contaminated water or food. Four sporozoites are contained inside each oocyst and are released by digestive processes in the intestinal epithelium. A schizogonic division takes place, resulting in the production of eight merozoites (type I merozoites), which reinvade new cells, and after a period of intracellular growth (type II merozoite), merozoites differentiate into micro and macrogametocytes that lead to fertilization and zygote formation. Mature zygotes develop into infective thin or thick-walled oocysts that are released from enterocytes. Infective thin-walled oocysts are broken in the intestine and lead to reinfections, while infective thick-walled oocysts are released into the environment through feces, contaminating water, soil, and food [38,39,40,41] (Figure 3).

Only nitazoxanide has demonstrated efficacy in human cryptosporidiosis. A number of new targets have been identified for chemotherapy, and progress has been made in developing drugs for these targets (Figure 4).

### 1.3. Malaria

Malaria is a parasitic disease considered a major public health problem because it causes a great number of morbidity and mortality cases, mostly in tropical and subtropical zones worldwide. In 2020, 241 million malaria cases were reported, and 627,000 deaths occurred, which represented a substantial increase compared to what was reported in 2019 [42]. Malaria is caused by *Plasmodium* parasites, which are intracellular Parasites transmitted mainly by the bite of female mosquitoes of the genus *Anopheles*. There are more than 120 *Plasmodium* species capable of infecting mammals, birds, and reptiles; nevertheless, only five species can infect humans, *P. malariae*, *P. falciparum, P. knowlesi, P. ovale*, and *P. vivax* [43,44].

In humans, parasites replicate asexually, while sexual reproduction takes place in *Anopheles* mosquitoes. Sporozoites injected by the *Anopheles* mosquito while feeding, reach the liver through the bloodstream and invade hepatocytes forming merozoites [45]. In the liver, *P. ovale* and *P. vivax* sporozoites can convert into hypnozoites, which are dormant forms that can relapse months or years later [44]. After liver parasite replication, merozoites are released into the bloodstream, and the intraerythrocytic cycle begins, in which rings, trophozoites, schizonts, merozoites, and gametocytes are developed [43]. Gametocytes are ingested by *Anopheles* mosquitos, and the cycle begins again. In the midgut of the mosquito, gametocytes develop a zygote, then a mobile ookinete capable of traversing the intestinal wall and forming an oocyst that, when mature, will develop sporozoites that will be released to invade the salivary glands [46,47]. During the intraerythrocytic cycle (Figure 5), the clinical features observed include high fever, chills, headache, myalgias, arthralgias, nausea, vomiting, and diarrhea [48,49]. *P. falciparum* infections can cause complicated malaria as a consequence of the cytoadherence phenomenon in which infected erythrocytes adhere to the vascular endothelium of different organs, causing cerebral malaria, acute respiratory distress syndrome, acute renal failure, anemia, thrombocytopenia, and placental malaria [48]. The intensity of clinical manifestation during complicated malaria varies according to age and the intensity of transmission, and if not treated promptly, mortality is high [40].

Multiple antimalarial drugs are used, including chloroquine, mefloquine, pyrimethamine, primaquine, and artemisinin derivatives [49] (Figure 6). Unfortunately, it is estimated that malaria morbidity and mortality have increased since 2020 due to the convergence of multiple factors, such as COVID-19 and Ebola outbreaks, natural disasters, and drug resistance, mainly to chloroquine and recently to artemisinin derivatives [44]. Malaria parasites have developed immune evasion strategies. Therefore, it is essential to find new alternatives for malaria control [42,44,50].

## 2. Antimicrobial Peptide Classification

The need to categorize everything that is known has facilitated the management of information in different settings, and chemical structures also have their own classification according to the functional groups present in their chemical structures. However, peptides are made up of a series of amino acids that are present in different functional groups depending on their biological activities. According to various authors, AMPs can be categorized according to different features, such as their charges (cationic, anionic), biological activities (antibacterial, antifungal, antiprotozoal, etc.), mechanisms of action, and even the source from which they were isolated (either from natural sources or synthetically) (Figure 7). A general form of classification is based on their physicochemical characteristics, which can be divided into four main groups: (1) α-helices, (2) β-pleated sheets, (3) those with mixed structures, and (4) those with atypical conformations [51,52,53]. The α helical structure is characterized by coiling on itself through peptide bonds and creating a type of tube. This conformation, in addition to providing amphipathic characteristics, allows it to be easily inserted into the cell membrane, creating channels [54]. β-pleated sheets are structures that fold back on themselves through N-H bonds of amino acids that conform by forming hydrogen bonds with the C=O groups of the opposite amino acids. Mixed structures can be present, within the same chain of amino acids, of the two conformations, both helical and β-pleated sheets. Finally, the atypical structures present forms that do not correspond to those mentioned above [55,56,57].

## 3. Mechanisms of Interaction by AMPs

Currently, research on AMPs has constantly been increasing, together with new research techniques such as bio-guided studies, in silico analysis, and synthesis, offering a broad number of peptides that have been described and evaluated in different biological models and clinical phases. To date, according to the Database of the Antimicrobial Activity and Structure of Peptides, 19,398 have been described, 82.5% of which are synthetic, and the rest have been isolated by natural sources, such as animals (75%), bacteria (12%), plants (9%) and fungi (4%) [58]. The knowledge of their mechanisms of action is continually increasing. It is noted that several peptides active against Apicomplexa parasites act directly on components of the cell membrane and extracellular components and the mechanism of surface membranes, mainly because AMPS are cationic and amphipathic molecules [59]. Most AMPs interfere with the correct functioning of the cytoplasmic membrane. With the progress in the discovery of AMPs and the elucidation of their mechanisms of action, researchers managed to understand different pathways by which they interact in both the host and host cells. Once the AMPs enter the cell, they can interact with components of the cytoplasm, altering the electrochemical balance as well as inhibiting metabolic processes essential for the survival of the parasite, altering cellular homeostasis and essential processes for cell replication [60].

AMPs’ mechanisms of action have been categorized into two main groups: those that exert a direct effect on killing cells and those that modulate the immune response. The first group is subdivided into two subgroups, those that kill directly by permeabilizing the cell membrane due to hydrophobic and electrostatic interactions of the peptides, and the second group, those peptides that kill by affecting the internal components of the cell acting as metabolic inhibitors [60,61,62,63].

## 4. Peptides Active against Apicomplexan Parasites

### 4.1. Toxoplasma gondii

Regarding AMPs that can modulate the immune response, it has been shown in vivo that HPRP-A1/A2 (amphipathic α-helical peptide) treatment induced a Th1/Tc1 response and elicited proinflammatory cytokines in mice infected with *T. gondii*; it is the only peptide with this type of mechanism of action in the parasite. These peptides affect the viability of tachyzoites at low concentrations; in addition, their activities against gram-negative and gram-positive bacteria and some pathogenic fungi have been reported [64]. A group of peptides that weaken the cell membrane, CA (2–8) M (1–18), lycosin-Ι, XYP1, XYP2, XYP3, longicin and longicin P4, have been tested in in vitro models against *T. gondii*. Lycosin-Ι was the most active, with an IC_50_ of 10 µM. However, other effects on the integrity of the tachyzoites were reported, such as the aggregation of the parasites induced by longicin P4, which in an in vivo model has managed to prolong the survival of mice for up to 11 days compared to the control [64,65,66,67,68,69,70].

Venoms from invertebrates such as spiders, scorpions, amphibians, and some reptiles are composed of different peptides, which in turn act mainly as modulators of ion channels and have been widely investigated in the pharmacological field for different diseases such as cancer and AIDS [71]. Some of these toxins have been evaluated against *Toxoplasma* [71]; however, peptides responsible for this activity have not been identified, although it is worth continuing with this research to identify the active peptides and elucidate their mechanisms of action. It should be noted that of the venoms and secretions evaluated, those obtained from the spiders *Ornitoctonus huwena* and *Chilobrachys jingzhao* were active against *T. gondii* tachyzoites at 3 µg/mL and increased the survival rate in vivo. There is only one study reporting peptide efficacy against *T. gondii* tissue cysts. The venom of the scorpion *Tityus serrulatus* was evaluated, and the Pep 1 peptide decreased the number of cerebral tissue cysts in infected mice, although its mechanism of action is still unknown [72,73,74].

Peptides with interesting biological activities have also been detected in marine organisms, as is the case of the conotoxin isolated from *Conus californicus* that affected tachyzoites in concentrations from 10 nM; of all the peptides investigated, it showed the highest activity [75].

Synthetic peptides represent an important component of known peptides to date, many of which have been identified from natural sources. Of the five synthetic peptides evaluated, Ac2-26 identified in human cells was able to reduce the parasite load from a concentration of 5 µM. (Table 1) [76].

### 4.2. Cryptosporidum spp.

AMPs that have been active in in vivo and in vitro evaluations against specific parasitic states of *Cryptosporidium* spp. are summarized in Table 2. Although human cryptosporidiosis is mainly caused by two species, *Cryptosporidium hominis* and *Cryptosporidium parvum*, AMP investigations against this parasite have specifically used *C. parvum* in both its sporozoite and oocyst forms and through evaluations in cell cultures and in vivo. The use of the meront phase has also been reported to determine the parasite load in these investigations. Approximately 16 cationic peptides have been tested to determine their anti-*Cryptosporidium* activity; three of them have been evaluated in more than one trial with similar results, and even combined treatments have been carried out to improve activity, as in the case of indolicidin, ranalexin, and magainin II. However, these combinations cannot be effectively compared because the pharmacological parameters of IC_50_ are not reported, and even in most of these evaluations, only 1 to 3 different concentrations up to 50 mM were evaluated. Evaluating these AMPs at different concentrations to determine their IC_50_ values, as well as their average cytotoxicity is of great importance to continue their research. Those with the best activity were the Buforin II and Magainin II peptides, which affected approximately 99.8% of the parasites in vitro at a concentration of 10 µg/mL [78,79,80,81,82,83,84,85]. However, the coupling of the peptide octarginine and the antibiotic nitazoxanide showed excellent results, lowering the IC_50_ value to 2.9 nM compared to the IC_50_ of nitozoanide alone, which was 197 nM. Of all the peptides evaluated, this combination showed the best results [86].

In in vivo experiments, peptides, such as glucagon-like peptides, in a treatment of 50 μg/kg of weight in calves infected with *C. parvum*, managed to reduce the symptoms of the infection, and eliminate the release of oocysts in the feces. Other peptides that act by regulating the immune response, SA35, and SA40, were isolated from *C. parvum*. These peptides were tested in mice infected and immunized with 5 μg of each peptide. Evaluations of the parasite load generate specific IgA antibodies and reductions of up to 96% of all intestinal forms of the parasite (Table 2) [87,88]. To date, the efficacies of none of these peptides have been demonstrated in clinical trials. However, it should not be ignored the biological activities that they present in low concentrations, and the synergistic effects that some reported peptides exert in combination with commercial antibiotics. The search for new alternatives for the treatment of cryptosporidiosis should focus on not only AMPs but also their combination with other active molecules, with the goal of attacking the parasite by different mechanisms of action.

**Table 2 antibiotics-11-01658-t002:** Synthetic AMPs with in vitro anti-*Cryptosporidium* activity.

AMP Name	Type	Evaluated Concentrations	Cytotoxicity	Activity and Possible Mechanisms of Action	
Buforin II [85]	α-Helical	20 μM	None in A549 cells.	Reduces sporozoites viability. Cell membrane Disruption	
Ranalexin [84]	Cationic	64 µg/mL	Non in A549 cells.	Sporozoites growth suppression. Cell membrane Disruption	
Ranalexin, Magainin II. Indolicidin [82]	Cationic, helix and tridecapeptide	50 mM	Non in A549 cells	Sporozoites growth suppression. Cell membrane damage by synergic effect between peptide and lipophilic antibiotics	
Shiva-10 [89]	Lytic peptide	10 µM	ND	Reduces sporozoite viability. Membrane lytic effect	
Cecropin P1, magainin II, ranalexin, and indolicidin [83]	Cationic peptides	50 μM	ND	Reduction in the proliferation of schizonts. Inhibition of Na/H and Na/Ca^2^ exchanges in the cell membrane	
KFFKFFKFF and IKFLKFLKFL [81]	Cationic peptides	100 µg/mL	ND	Reduction in the viability of sporozoites. Cell membrane disruption	
SMAP-29, BMAP-28, PG-1, Bac-7 [80]	Helical peptides	100 μg/mL	ND	Strong cytotoxic effect on sporozoites. Alterations in the glycoprotein of the apical complex	
Indolicidin, Magainin II, Ranalexin [79]	Cathionic peptides	50 μM	ND	Reduction in merozoites proliferation	
Octaarginine-6-FAM-Nitazoanide combination [86]	Cathionic peptides	197 nM	No cytotoxic effects in human ileocecal adenocarcinoma cells	Reduction in trophozoites and meronts replication	
Lactoferrin B, cathelicidin LL3, indolicidin, βdefens1in, ß defensin 2. [90]	Cathionic peptides		10 µg/mL	Low cytotoxic effect in human colorectal adenocarcinoma cells	Inhibition of sporozoites attachment and invasion. Transmembrane pore formation
Buforin II, Magainin II, Lasalocid. [78]	Cathionic peptides		10 µg/mL	ND	Reduction in oocysts infectivity. Membrane disruption

ND: Not determined. IC_50_ values were not established.

### 4.3. Peptides Active against Plasmodium spp.

Peptides against *Plasmodium* have multiple mechanisms of action that cause decreases in parasitemia (Table 3), and one of the predominant mechanisms is the interaction of peptides with enzymes causing their inhibition and, consequently damage to the metabolic pathways in which they participate. For example, for the maintenance and processing of genetic material, peptides can inhibit the enzyme purine nucleoside phosphorylase of *P. falciparum*, and the enzyme dihydrofolate reductase-thymidylate synthase, resulting in the death of the parasite [91,92]. Another example of enzyme inhibition occurs during the erythrocyte cycle, during the digestion of hemoglobin in the digestive vacuole for protein biosynthesis and heme crystallization, a process that is catalyzed by enzymes such as falcispainins that, if their function is inhibited, the parasite cannot obtain the amino acids necessary for protein synthesis and therefore would die; this strategy is used by certain peptides, such as CYS-IHL and CYS-cIHL, that are capable of inhibiting these enzymes [93,94].

Peptide–membrane interactions and H+ homeostasis disruption in *P. falciparum* asexual blood stages

Other targets of peptides are proteins and membranes, which, if damaged, can modify the morphology of the parasite; however, not all peptides have parasiticidal effect, and some only stop the development of *Plasmodium* spp., which is reflected in the slowed kinetics of the life cycle [92,94,95,96,97,98].

In addition to reducing parasitemia, some peptides are capable of modifying the immune response in the host by reducing the overproduction of proinflammatory cytokines and, as a consequence, modulating damage to organs that are severely affected, such as the liver [99].

Nevertheless, more information is needed to elucidate the mechanisms of action of antimicrobial peptides against *Plasmodium* spp.

## 5. Concluding Remarks and Future Research Directions

Antimicrobial peptides have been described in many species, including fungi, plants, insects, and humans (allowing access to an endless number of possible peptides with diverse biological activities), and are currently presented as a therapeutic solution to control different pathogenic microorganisms. Microorganisms that cause diseases in humans are constantly evolving, which represents a challenge in the pursuit of effective treatments against these pathogens. Some characteristics that make peptides attractive as potential drugs are that they have been evolving for almost the same amount of time as the species that produce them, and their effects on the control of microorganisms are very remarkable. Some peptides are being used in experimental phases, and others are already marketed, e.g., peptides against fungal agents such as *Candida albicans*, *Cryptococcus neoformans*, and *Fusarium oxysporum*. Some peptides have been developed for topical application against human papilloma virus, and others have been developed against protozoa and nematodes, gram-negative bacteria, tumors, and as neuroprotectors.

Endogenous bioactive peptides can be produced in different cell types, such as neural cells, immune cells, or glands, while exogenous peptides can be obtained from nutrients, insects, nematodes, or marine organisms. Cecropin is one of the most explored insect peptides that can destroy cell membranes and inhibit proline uptake.

Unlike other parasites, Apicomplexans have complex life cycles comprised of different stages characterized by rapid replication, which enables adaptation to drug treatment. The Apicomplexa invasion process involves secretory organelles housing proteins that allow host-cell entrance and the development of an intracellular compartment in which the parasites reproduce asexually. As intracellular organisms, their nutritional needs rely on biosynthetic pathways or salvaging metabolites from their host [103]. Apicomplexa drug targets include calcium-dependent protein kinases, mitochondrial electron transport chain, proteins secretion pathways, type II fatty acid synthesis, DNA synthesis and replication, and, DNA expression, among others [104]. Most of the peptides reviewed in this text produce the disruption of parasite cell membranes, in contrast to conventional chemotherapeutic drugs, which act on precise targets such as DNA or specific enzymes. Nonetheless, plasma membrane disruption, produces fast depolarization triggering protein and DNA/RNA inhibition synthesis, which can lead to parasite death. Some peptides are rich in amino acids, such as tryptophan and lysine, that might have an effect on anionic biological membranes, producing pores, which allow peptides to distribute into internal membranes and organelles [64].

Unlike Apicomplexan, hemoflagellate protozoa, such as *Trypanosoma* and *Leishmania*, have less complex life cycles. Various research groups have been dedicated to the discovery and structural elucidation of novel peptides against these parasites since the early 90s. Extracellular forms (promastigote and trypomastigote) are the most common stages used for the screening of peptides’ activity [105]. The antiprotozoal activity is supposed to occur by membrane disruption, apoptosis, or by immunomodulatory responses. In vivo assessments are considerably underexplored, due to their rapid degradation by endogenous proteases [105]. It seems that peptide-based antiprotozoal drug development, presents several challenges related to the complex life cycles. Therefore, computational models and tools for the prediction of peptide activity are urgently needed. However, peptides have some advantages over traditional drugs, such as slower emergence of resistance [106].

There are some issues to consider while scaling up peptide design. Peptides have various limitations that could hinder their anti-Apicomplexa therapeutic use. They have unfavorable plasma stability, are unable to cross the cell membrane to target intracellular targets, degrade easily, and have poor penetration of the intestinal mucosa; thus, it can be assumed that they are not good candidates to treat intracellular parasites. [107]. Nonetheless, the results obtained so far show that they can be a good alternative to control these parasites. It must be taken into consideration, that novel peptides must easily reach intracellular targets with little or no toxicity to mammalian cells. To improve these disadvantages, encapsulation into a micro- or nanoparticle, can be achieved, as well as in silico sequence-based prediction of cell-penetrating and toxicity. Penetratin-like peptides bind to glycosaminoglycans at the cell surface. Natural DNA-binding peptides can be the source for designing cell-penetrating peptides, such as those rich in lysine, or arginine [107].

Although there are currently some pharmacological alternatives for the control of Apicomplexan parasites, these are sometimes inefficient, especially due to resistance mechanisms and severe side effects, and they do not act against all parasite stages and sometimes restrict access to some intracellular locations. Based upon the abovementioned results, it seems that synthetic peptides, as well as those derived from natural sources, could be promising alternatives for the treatment of infectious diseases. It is necessary to develop new anti-Apicomplexan compounds combining drug research pathways, such as in silico rational drug design and bio-guided natural substance studies, to identify new molecules that might be able to act directly in the parasites or indirectly by activating the host immune system.

As reported in the literature, peptides show a broad antimicrobial spectrum; therefore, it would be recommended to explore their synergistic ability in combination with those drugs in which resistance is reported, their capacity to decrease or increase the adverse effects of currently used drugs, and their distribution in the parasite and in the host cell. Genetic engineering or chemical modification of these peptides to improve their functional properties would also be recommended. There is a high potential for the use of antimicrobial peptides, and more research in this field can lead to promising results that can have considerable effects on the control of human Apicomplexan parasites.

## Figures and Tables

**Figure 1 antibiotics-11-01658-f001:**
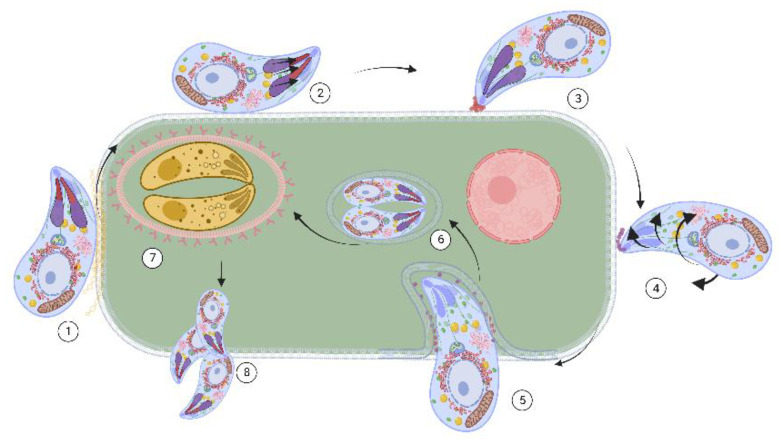
Active invasion of *T. gondii*. In Apicomplexan, three types of secretory organelles are observed: micronemes, rhoptries, and dense granules, carrying characteristic proteins. Attachment to host cell membrane via micronemes (MIC) proteins (**1**). Invasion and moving junction development by secretion of proteins from rhoptries neck (RON) and rhoptires (ROP) (**2**,**3**). Internalization via secretion of RON/AMA proteins (**4**). Parasitophorous vacuole development via granule dense proteins (GRA) (**5**). Proliferation and tachyzoite asexual replication (**6**). Increases immune response, interconversion to bradyzoite, and tissue cyst formation (**7**). Decreases immune response, interconversion to bradyzoites-tachyzoites, and dissemination of the parasite (**8**). Tachyzoites cause acute infection, leading to severe toxoplasmosis. While several drugs are available against tachyzoites, there is no treatment against tissue cysts, which are responsible for chronic infection. An ideal anti-*Toxoplasma* drug should be effective against both stages and prevent interconversion. Protein targeting secretory organelles is a matter of interest. Created with BioRender.com under license to publish by Anacleto SJ.

**Figure 2 antibiotics-11-01658-f002:**
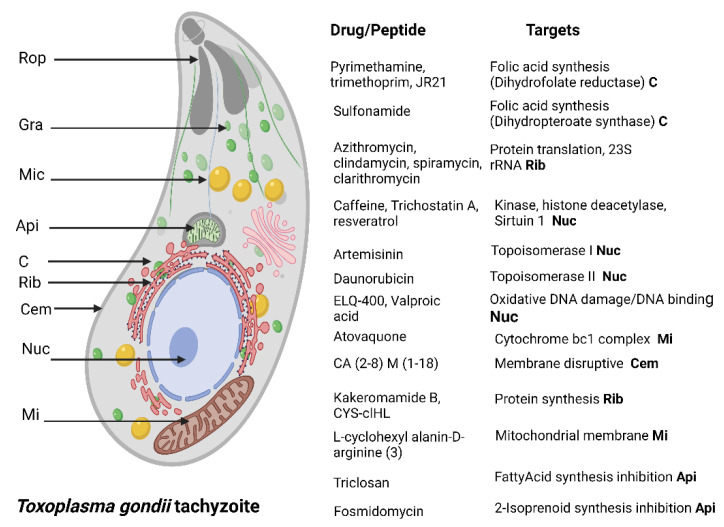
*T. gondii* tachyzoite drug targets. Rop, rohptry. Gra, dense granule. Mic micronemes. Api, apicoplast. C, cytoplasm. Cem, cell membrane. Rib, ribosome. Nuc, nucleus. Mi, mitochondrion. Created with BioRender.com under license to publish by Anacleto SJ.

**Figure 3 antibiotics-11-01658-f003:**
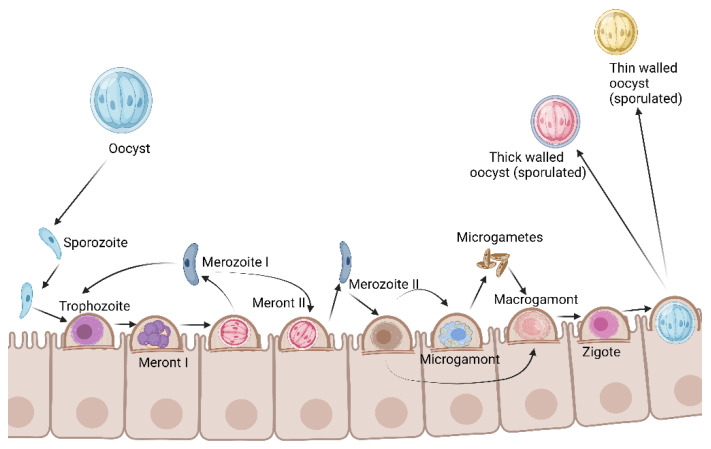
*Cryptosporidium* spp. development in the host cell. Anti-cryptosporidial drug development challenges a major problem: the discovery of systemic drugs that can reach epicellular parasites (preventing schizogonic reproduction); and the absorption by patients undergoing diarrhea. Created with BioRender.com under license to publish by Anacleto SJ.

**Figure 4 antibiotics-11-01658-f004:**
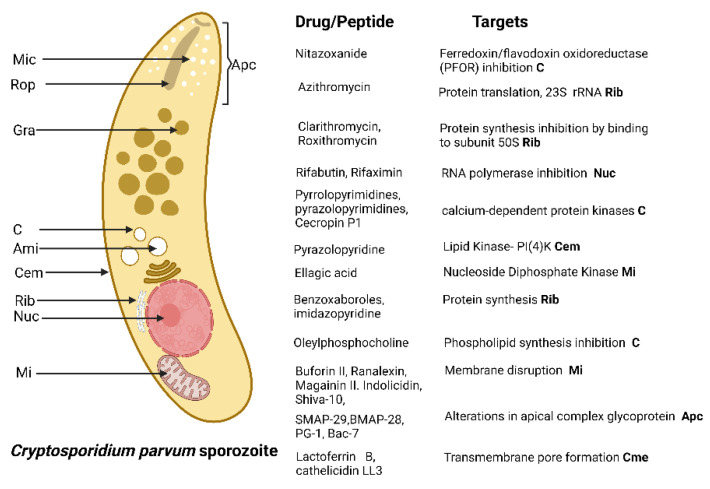
*Cryptosporidium* drug targets. *Cryptosporodium* lacks many drug targets present in other Apicomplexans because of a simplified metabolism and the absence of de novo nutrient synthetic pathways. Mic micronemes. Rop, rohptry. Gra, dense ganule. Apc, apical complex. C, cytoplasm. Ami, amylopectin granules. Cem, cell membrane. Rib, ribosome. Nuc, nucleus. Mi, mitochondrion. Created with BioRender.com under license to publish by Anacleto SJ.

**Figure 5 antibiotics-11-01658-f005:**
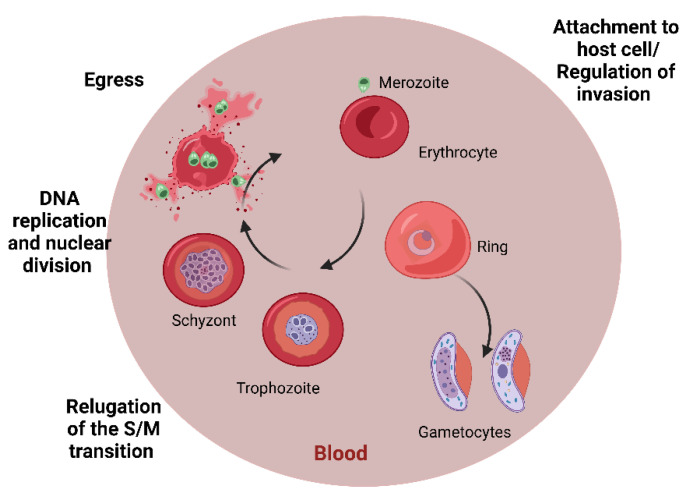
*Plasmodium* spp. intraerythrocytic cycle. Most antimalarial drugs target the asexual erythrocytic stages (rings, throphozoites, and schyzonts).

**Figure 6 antibiotics-11-01658-f006:**
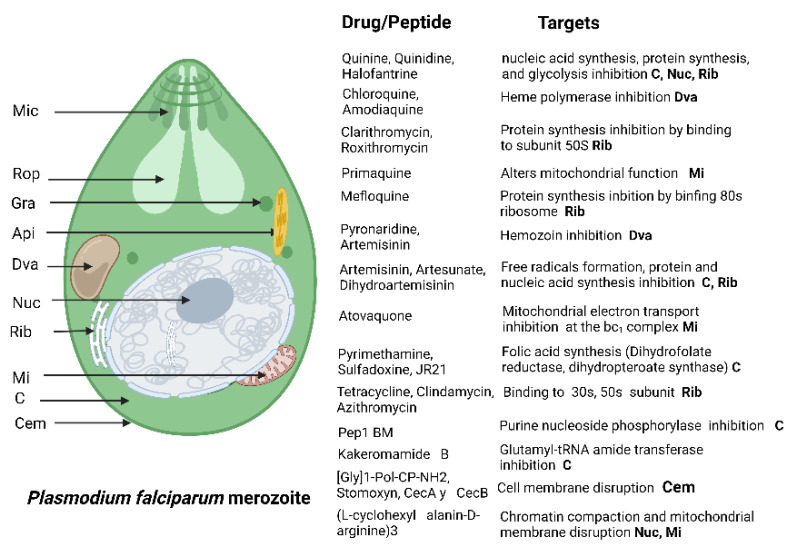
*Plasmodium* spp. drug targets. Antimalarial drugs such as aryl amino alcohol (chloroquine, mefloquine, primaquine), antifolate compounds (pyrimethamine), and artemisinin derivatives (artesunate, artemether) target the asexual erythrocytic stages of the parasite. Mic micronemes. Rop, rohptry. Gra, dense ganule. Api, apicoplast. Dva, digestive vacuole. Nuc, nucleus. Rib, ribosome. Mi, mitochondrion. C, cytoplasm. Cem, cell membrane. Created with BioRender.com under license to publish by Anacleto SJ.

**Figure 7 antibiotics-11-01658-f007:**
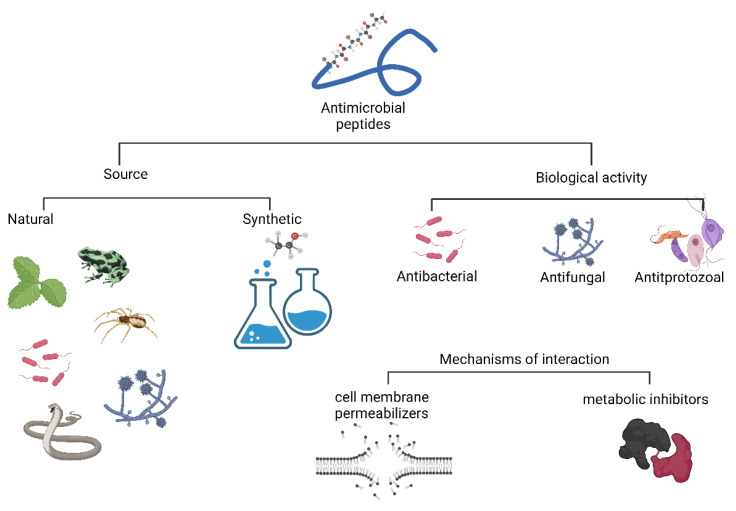
Antimicrobial peptides classification and interaction.

**Table 1 antibiotics-11-01658-t001:** AMPs with in vitro anti-*Toxoplasma* activity on tachyzoites.

AMP Name	Type	Source	Evaluated Concentrations	Cytotoxicity	Activity and Possible Mechanism of Action	IC_50_
Frog skin secretion [71]	ND	*Phyllomedusa distincta* [Amphibia] *Corythomanti greening* [Amphibia]	25 µg/mL and 22 µg/mL respectively	None in human Fibroblasts	Inhibits invasion	ND
CA (2–8) M(1–18) [65]	Cecropin/ melittin hybrid peptide	Synthetic	5 µM	None in human fibroblasts	Reduces viability Membrane lytic activity	ND
Ac2-26 peptide mimetic of Annexin A1 [76]	Human peptide	Synthetic	5 μM	ND	Decreases proliferation rate	ND
Lycosin-Ι [68]	Linear peptide	*Lycosa singoriensis* [Arachnida]	20 µM	Cytotoxic at 34.69 µM in human fibroblasts	Invasion and proliferation inhibition. Cell membrane alteration	28 and 10.08 μM for intracellular and extracellular tachyzoites, respectively
Longicin [69]	Cationic	*Haemaphysalis longicornis* [Arachnida]	50 µM	ND	Reduces proliferation. Cell membrane disruption	ND
ND [72]	Venoms	*Ornitoctonus huwena Chilobrachys jingzhao* [Arachnida]	12.5 µg/mL	Cytotoxic to Hella cells	Proliferation and invasion reduction	ND
XYP1 [67]	Cationic	synthesized	2.5–40 µM	Low cytotoxicity at 20 µM in human fibroblasts	Inhibition of viability, invasion, and proliferation. Damage to membrane associated proteins (HSP29)	38.79 µM
cal14.1a [75]	Conotoxin	*Conus* *californicus* [Gastropoda]	10–50 µM	Not detected up to 50 µM in Hep-2 cells	Affects viability and replication by disrupting cell membrane	ND
ND [73]	Venom	*Hemiscorpius Lepturus* [Arachnida]	50 µg/mL	CC_50_ 72.46 µg/mL (Vero cells)	Reduces viability and invasion. Probably damaging ion channels and enzymatic activity	39.06 µg/mL
Killer peptide (KP) [77]	Decapeptide	Synthetic	25–200 µg/mL	Nontoxic to Vero cells. Genotoxic effects were reported	Reduces invasion and proliferation. Maybe triggers an apoptosis like cell death	ND
Longicin P4 [70]	ND	*Haemaphysalis Longicornis* [Arachnida]	50 µM	Nontoxic up to 25 µM	Reduces proliferation. Induces aggregation and affects membrane integrity	ND
HPRP-A1/A2 [64]	Cationic peptide	Synthetic	10–40 µg/mL	Nontoxic in peritoneal macrophages.	Reduces viability, adhesion, and invasion	ND
Sub6-B, Pep1, Pep2a and Pep2b [74]	Venom fractions	*Tityus* *serrulatus* [Arachnida]	100 µg/mL	Nontoxic in peritoneal macrophages	Reduces invasion and replication. Disruption of cell membrane	ND

ND: Not Determined.

**Table 3 antibiotics-11-01658-t003:** AMPs with in vitro anti-Malarial activity.

AMP Name	Type	Source	Evaluated Concentration	Cytotoxicity	Activity and Possible Mechanisms of Action	IC_50_
Pep1 BM [91]	ND	Synthetic	20 µL	ND	Inhibition of purine nucleoside phosphorylase in *P. falciparum* rings	16.14 μg/mL
JR21 [92]	ND	Synthetic	10 µM	ND	Dihydrofolate reductase- thymidylate synthase inhibition in *P. falciparum* rings	3.87 µM
CYS-IHL [94]	Linear	Synthetic	69.91 µM	Noncytotoxic in human liver carcinoma cell.	Hemoglobinase activity inhibition in late *P. falciparum* Trophozoites	27.55 µM
Kakeromamide B [95]	Cyclic	*Moorea* *producens* [Cyanobacteria]	11 µM	Noncytotoxic in HEK293T and HepG2 cells	Reduction in proliferation of *P. falciparum* sexual blood-stages and *P. berghei* liver-stage. High affinity to actin, sortilin and subunit A of glutamyl-tRNA amide transferase	8.9 µM
[Gly]1-Pol-CP-NH2 [96]	ND	Synthetic derived from Pol-CP-NH2	6.25 µM	Cytotoxic in human mammary adenocarcinoma, Hep G2, SHSY-5Y, and SK-mel-147	Cell membrane disruption in *P. falciparum* sporozoites	ND
Crotamine [97,98]	Cationic	*Crotalusdurissusterrificus* [Lepidosauria]	20 µM	No hemolytic activity in human erythrocytes	Peptide–membrane interactions and H+ homeostasis disruption in *P. falciparum* asexual blood stages	1.87 µM
(L-cyclohexyl alanin-D- arginine) 3 [99]	ND	Synthetic	59.16 ng/mL	No cytotoxic effects in human erythrocytes and leukocytes	Chromatin compaction and mitochondrial membrane disruption in *P. falciparum* asexual blood stages	8.94 ng/mL
rR8-JR21 [92]	ND	Synthetic	13.22	ND	Dihydrofolate reductase-thymidylate synthase inhibition in *P. falciparum* ring stages	1.53 µM
LZ1 [100]	Linear peptide	Synthetic derived fromcathelicidin-BF	25 µM and 4 mg/kg	ND	Blockade of ATP production by selective inhibition of pyruvate kinase activity in *P. falciparum* blood stages.	3.045 µM
Mtk-1 y Mtk-2 [101]	Rich in proline	*Drosophila melanogaster* [Insecta]	50 µM	Hemolytic activity in pig and mouse (CD1) erythrocytes	Cell membrane disruption in *P. falciparum* asexual blood stages	ND
Stomoxyn [101]	ND	*Lucilia sericata* [Insecta]	50 µM	Hemolytic activity in highest concentrations in pig and mouse (CD1) erythrocytes	Cell membrane disruption in *P. falciparum* asexual blood stages	ND
CecA y CecB [101]	Linear cations	*Galleria mellonella* [Insecta]	50 µM	Hemolytic activity in highest concentrations in pig and mouse (CD1) erythrocytes	Cell membrane disruption in *P. falciparum* asexual blood stages.	ND
[Arg]3-VmCT1-NH2, [Arg]7-VmCT1-NH2 [102]		Synthetic	5 µM/L	Lower Cytotoxic effects in MCF-7 human breast epithelial cells, CC_50_ 20 and 18 µM/L	Cell membrane disruption in *P. gallinaceum* sporozoites	0.57, 0.51 µM/L
VmCT1-NH2 [102]		*Vaejovis* *mexicanus* [Arachnida]	5 µM/L	CC_50_ 8.3 µM/L in MCF-7 human breast epithelial cells	Cell membrane disruption in *P. gallinaceum* sporozoites	0.49 µM/L

ND: not determined.

## Data Availability

Not applicable.

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
