# Peer review of "Bioactive Peptides against Human Apicomplexan Parasites"

_antibiotics, 2022, doi:10.3390/antibiotics11111658_

Round 1

Reviewer 1 Report

The work is a review of antimicrobial peptides with potential effect on three Apicomplexan parasites affecting humans, Plasmodium, Toxoplasma and Cryptosporidium. Currently drugs have some or no efficacy, hence, the importance of seeking alternatives for their treatment and/or control of these parasitic diseases.

1     1.  Observations that need to be revised and corrected are highlighted in yellow along the manuscript (attached in pdf). English syntax, wording,  paragraphs that are difficult to understand. Every scientific name should be written in italics. Please check them all.

3.    2.  Figures. The figures provide no new information and can be found in a classic parasitology textbook. To date, there are no significant changes in the knowledge of life cycles of the three parasites. Legends are too small, difficult to read without effort. Suggestion to the authors: modify the three figures, but instead of explaining the development of each life cycle, incorporate in the parasitic stages the therapeutic targets for AMP. Be creative…

4.       3. The most important observations concern the tables.  Too much information which makes it difficult to follow and loses the importance of the review. Complete paragraphs with connectors, repetition of words in each row that could be put together by combining the rows. In some cases AMPs are detailed but generalized in others. The source of the AMPs should include the taxonomic group (class/order). Too much detail in the activity column with a mixture of in vitro activity, evolutionary stages, immune response, AMP types. Difficult to follow. Suggestion to authors: redo the tables. It would be more understandable if instead of making a table per parasite, per type of activity. Please try to summarize the AMPs activity in another way. Avoid connectors, comparisons with control groups, just write the concentration with maximum activity. Incorporate new columns to avoid text overload. Particularly noteworthy is the inclusion of the SA35 and SA40 antigens in table 2. Unless the authors have a good explanation for including them there, it should be removed. By including antigens, the authors are losing the division between antimicrobial and immunogenic peptides.

5.      4.  Compare the use of AMPs with other parasites, e.g. Leishmania, Trypanosoma. More discussion is needed along the manuscript on therapeutic targets and sources of origin, beyond the three extensive tables.

6.     5.  Remove in the title “Novel Approach”.

7.      6.  Review references and correct them according to MDPI standards.

Author Response

  1. Observations that need to be revised and corrected are highlighted in yellow along the manuscript (attached in pdf). English syntax, wording, paragraphs that are difficult to understand. Every scientific name should be written in italics. Please check them all.

The manuscript was edited by a native speaker expert on the subject in the Elsevier web shop, the edition certificate has been sent to the editor. Some paragraphs have been changed based on the PDF notes, some others were not changed because the other reviewers found them appropriate. All Scientific names are now in italics.  Changes are highlighted in yellow.

  1. Figures. The figures provide no new information and can be found in a classic parasitology textbook. To date, there are no significant changes in the knowledge of life cycles of the three parasites. Legends are too small, difficult to read without effort. Suggestion to the authors: modify the three figures, but instead of explaining the development of each life cycle, incorporate in the parasitic stages the therapeutic targets for AMP. Be creative…

The figures of the biological cycles were replaced by others where targets and parasites stages are shown. Figure legends got bigger.

  1. The most important observations concern the tables. Too much information which makes it difficult to follow and loses the importance of the review. Complete paragraphs with connectors, repetition of words in each row that could be put together by combining the rows. In some cases AMPs are detailed but generalized in others. The source of the AMPs should include the taxonomic group (class/order). Too much detail in the activity column with a mixture of in vitro activity, evolutionary stages, immune response, AMP types. Difficult to follow. Suggestion to authors: redo the tables. It would be more understandable if instead of making a table per parasite, per type of activity. Please try to summarize the AMPs activity in another way. Avoid connectors, comparisons with control groups, just write the concentration with maximum activity. Incorporate new columns to avoid text overload. Particularly noteworthy is the inclusion of the SA35 and SA40 antigens in table 2. Unless the authors have a good explanation for including them there, it should be removed. By including antigens, the authors are losing the division between antimicrobial and immunogenic peptides.

The tables have been completely modified, now they include concise and precise information, as well as the changes suggested by the reviewers.

  1. Compare the use of AMPs with other parasites, e.g. Leishmania, Trypanosoma. More discussion is needed along the manuscript on therapeutic targets and sources of origin, beyond the three extensive tables.

More discussion about other parasites, therapeutic targets and sources of origin have been added.

  1. Remove in the title “Novel Approach”.

“Novel Approach” was eliminated from the title.

  1. Review references and correct them according to MDPI standards.

References have been corrected according to MDPI standards.

Reviewer 2 Report

This review focus on the research work on the discovery of peptides the biological activity against apicomplexa organisms.

Then are some issues I feel that need revision. The title " Novel approaches..." make the reader expect that the review maybe about novel technologic or system for peptides identification. But is not.

Plasmodium falciparum is included on the article body but not included on the abstract.

The structure of including parasites' transmission origin figures before their life cycle is strange.

Apicomplexa parasites are characterized by its special organelle, apicoplast. The review neglects work performed in peptides to block parasites invasion which is an important and large field of research.

Author Response

  1. Then are some issues I feel that need revision. The title " Novel approaches..." make the reader expect that the review maybe about novel technologic or system for peptides identification. But is not.

      Novel Approach was eliminated from the title.

  1. Plasmodium falciparum is included on the article body but not included on the abstract.
  2. P. falciparum was included in the abstract.
  3. The structure of including parasites' transmission origin figures before their life cycle is strange.

These figures were eliminated, different figures were designed to show the therapeutic targets.

  1. Apicomplexa parasites are characterized by its special organelle, apicoplast. The review neglects work performed in peptides to block parasites invasion which is an important and large field of research.

It is specified in the tables if the peptides act on the viability, invasion, or replication of the parasite.  In the manuscript, changes are highlighted in yellow .

Reviewer 3 Report

The authors of the review article “Novel Approach to Bioactive Peptides Against Human Apicomplexan Parasites” reviewed three important parasitic diseases namely malaria, toxoplasmosis, and cryptosporidiosis caused by apicomplexan parasites and their novel therapeutic approaches/strategies using bioactive peptides. The authors summarized recent updates on peptides active against those three important apicomplexan diseases and their mode of action against the parasites. This review might provide quick updates on the topics to its reader and might have great value in its specific field. However, the authors may improve the review further by considering the following comments:

1.  Authors provided an orientation about the apicomplexan diseases in the abstract and introduction section where they mentioned about malaria first followed by toxoplasmosis and cryptosporidiosis. However, while they described those diseases they described toxoplasmosis first followed by cryptosporidiosis, and finally malaria. It might be better to follow the same order in both cases for better orientation to help its reader.

2.  Authors classified AMPs and their mechanism of interaction in section-2 and 3. It might be better to have a quick look by its reader if the authors summarize the information (in section-2, and 3) in a flow diagram.

3.  It might be better if the authors could add/mention specific molecular targets of each AMP in the parasites (in the case where information are available) and their possible toxicities (cytotoxicity on host cells) in Table 1, 2, and 3.

4. Though mentioned shortly (within the review) about the limitations of using AMP as therapeutic agents, it might be better if the authors describe this issue somewhat elaborately in the “Concluding remarks and future research direction section (section 5) as well.

 5. It is better to explain in short, the mechanism of the current treatment for toxoplasmosis, cryptosporidiosis, and malaria.

5.  Line 173: Not C. parvum it should be Cryptosporidium parvum because you have used it the first time.

6.  Line 242: “Apicomplexan parasites” are a more general name than apicomplexan microorganisms.

7. Line 385: Table 1. looks not systematic, activity and mode of action (mechanisms) should be separated.

8. Line 398: IC50 or IC50? It is better to use the same format for the whole manuscript.

9. Line 73: in vitro should be in Italic font.

10. Line 141: in Figure 1. The authors mentioned that T. gondii is transmitted to humans by drinking unpasteurized goat milk. Unpasteurized milk is the infection source but not only goat milk, it should be summarized more in general.

Author Response

  1. Authors provided an orientation about the apicomplexan diseases in the abstract and introduction section where they mentioned about malaria first followed by toxoplasmosis and cryptosporidiosis. However, while they described those diseases they described toxoplasmosis first followed by cryptosporidiosis, and finally malaria. It might be better to follow the same order in both cases for better orientation to help its reader.

The order of diseases was unified.

  1. Authors classified AMPs and their mechanism of interaction in section-2 and 3. It might be better to have a quick look by its reader if the authors summarize the information (in section-2, and 3) in a flow diagram.

A flow diagram has been added.

  1. It might be better if the authors could add/mention specific molecular targets of each AMP in the parasites (in the case where information are available) and their possible toxicities (cytotoxicity on host cells) in Table 1, 2, and 3.

Specific molecular target has been added to the tables as well as cytotoxicity.

  1. Though mentioned shortly (within the review) about the limitations of using AMP as therapeutic agents, it might be better if the authors describe this issue somewhat elaborately in the “Concluding remarks and future research direction section (section 5) as well.

More description has been added.

  1. It is better to explain in short, the mechanism of the current treatment for toxoplasmosis, cryptosporidiosis, and malaria.

Mechanisms of action of current treatments were added.

  1. Line 173: Not C. parvum it should be Cryptosporidium parvum because you have used it the first time.

This suggestion was added.

  1. Line 242: “Apicomplexan parasites” are a more general name than apicomplexan microorganisms.

This suggestion was added.

  1. Line 385: Table 1. looks not systematic, activity and mode of action (mechanisms) should be separated.

Separation was made

  1. Line 398: IC50 or IC50? It is better to use the same format for the whole manuscript.

Format was modified.

  1. Line 73: in vitro should be in Italic font.

In vitro was modified

  1. Line 141: in Figure 1. The authors mentioned that T. gondii is transmitted to humans by drinking unpasteurized goat milk. Unpasteurized milk is the infection source but not only goat milk, it should be summarized more in general.

This figure was eliminated, as different figures were design.

Reviewer 4 Report

The manuscript entitled “Novel approach to bioactive peptides against human Apicomplexan parasites” by Rivera-Fernández et al. is a review on what is known regarding antimicrobial peptides tested against the three apicomplexan parasites Toxoplasma, Cryptosporidium and Plasmodium. The abstract states that a review regarding the experimental effects of bioactive peptides will be given. 

In an age where a vast amount of information is put out into the universe I believe writing constructive review articles are quit daunting. In light of this I would like to make the following comments regarding the manuscript:

In general the manuscript describes the life cycle of all three parasite’s of interest. Although this information sketches the complexity of the life-cycles of these organisms and why it is so difficult to target them with antimicrobial drugs, I do feel that the rest of the information given is not really discussed in the context of the life-cycles presented. The question therefore arises what the need is to know what the life-cycle entails in combination with the information summarised for all the peptides tested against these organism. Therefore either the two sets of information should be linked to give more context, or the inclusion of the life-cycles should be reconsidered.

This bring me to the second part of the manuscript that deals with the peptide information currently known. Currently, the peptides tested and activities obtained are reviewed and summarised. Although this information is handy to get an overview of peptides investigated in the past years, the potential of these compounds in the context of scalability, toxicity towards the human host and the potential of antimicrobial resistance is not explored. To really assess the potential of peptides as antiparasitics, these aspects for the peptides discussed must be included. As the manuscript stands the information given leaves the reader with the sense of “so what?”, which, coming back to my initial statement that as vast amounts of information is being published, questions the sensibility and usefulness of the review. This also leads me back to the title of the review which states the “novel approach”. However, the approach and the novelty of the approach is not set out clear enough in the body of work. Due to these factors, I fell the review needs to be reworked to make the novelty and actual practical use of peptides as antiparasitic agents more clear. I feel as the manuscripts stands at the moment it is not ready for publication, yet.

Minor details that can be addressed is given in the annotated pdf included in the review. Two main issues with presentation needs to be addressed.

1.     Biorender is an amazing tool to make very nice illustrations and although visuals can explain text more effectively, I think the two figures on the transmissions of the parasites are redundant.

2.     The tables need to be simplified to be more user friendly. At the moment they are quite tedious to follow.

Author Response

  1. In general the manuscript describes the life cycle of all three parasite’s of interest. Although this information sketches the complexity of the life-cycles of these organisms and why it is so difficult to target them with antimicrobial drugs, I do feel that the rest of the information given is not really discussed in the context of the life-cycles presented. The question therefore arises what the need is to know what the life-cycle entails in combination with the information summarised for all the peptides tested against these organism. Therefore either the two sets of information should be linked to give more context, or the inclusion of the life-cycles should be reconsidered.

More context was added.

  1. This bring me to the second part of the manuscript that deals with the peptide information currently known. Currently, the peptides tested and activities obtained are reviewed and summarised. Although this information is handy to get an overview of peptides investigated in the past years, the potential of these compounds in the context of scalability, toxicity towards the human host and the potential of antimicrobial resistance is not explored. To really assess the potential of peptides as antiparasitics, these aspects for the peptides discussed must be included. As the manuscript stands the information given leaves the reader with the sense of “so what?”, which, coming back to my initial statement that as vast amounts of information is being published, questions the sensibility and usefulness of the review. This also leads me back to the title of the review which states the “novel approach”. However, the approach and the novelty of the approach is not set out clear enough in the body of work. Due to these factors, I fell the review needs to be reworked to make the novelty and actual practical use of peptides as antiparasitic agents more clear. I feel as the manuscripts stands at the moment it is not ready for publication, yet.

More information regarding scalability, toxicity and the potential of antimicrobial resistance was included. “Novel approach” was eliminated from the title.

  1. Minor details that can be addressed is given in the annotated pdf included in the review. Two main issues with presentation needs to be addressed.

The comments in the pdf were taken into consideration and the changes were made.

  1.  Biorender is an amazing tool to make very nice illustrations and although visuals can explain text more effectively, I think the two figures on the transmissions of the parasites are redundant.

Transmission figures were eliminated.

  1. The tables need to be simplified to be more user friendly. At the moment they are quite tedious to follow.

Tables were modified.

ALL CHANGES IN THE MANUSCRIPT ARE HIGHLIGHTED IN YELLOW.

Round 2

Reviewer 1 Report

The authors corrected the remarks made in the first round and the manuscript was substantially improved.

Just a single comment. The authors mentioned in the title and objectives  "human Apicomplexans parasites" but the subtitles refered to the diseases caused by these parasites. I suggest to put the name of the parasites and not the name of the diseases.